# Literary runaway: Increasingly more references cited per academic research article from 1980 to 2019

**Can Dai**[1,2], **Quan Chen**[1], **Tao Wan**[3], **Fan Liu**[4], **Yanbing Gong**[5], **Qingfeng Wang**[4]*

**1** School of Resources and Environmental Science, Hubei University, Wuhan, China, **2** Hubei Key Laboratory of Regional Development and Environmental Response, Hubei University, Wuhan, China, **3** Fairy Lake Botanical Garden, Shenzhen, China, **4** Key Laboratory of Aquatic Botany and Watershed Ecology, Wuhan Botanical Garden, Chinese Academy of Sciences, Wuhan, China, **5** State Key Laboratory of Hybrid Rice, College of Life Sciences, Wuhan University, Wuhan, China

* qfwang@wbgcas.cn

**Data Availability Statement:** Data and R codes are held by Figshare Digital Repository. DOI: 10.6084/m9.figshare.14811363.

## Abstract

References are employed in most academic research papers to give credits and to reflect scholarliness. With the upsurge in academic publications in recent decades, we are curious to know how the number of references cited per research article has changed across different disciplines over that time. The results of our study showed significant linear growth in reference density in eight disciplinary categories between 1980 and 2019 indexed in Web of Science. It appears that reference saturation is not yet in sight. Overall, the general increase in the number of publications and the advanced accessibility of the Internet and digitized documents may have promoted the growth in references in certain fields. However, the seemingly runaway tendency should be well appreciated and objectively assessed. We suggest that authors focus on their research itself rather than on political considerations during the process of writing, especially the selection of important references to cite.

## Introduction

Science is universally documented and communicated among researchers by means of papers published in academic journals. The most prominent difference between academic papers and their popular counterparts is, perhaps, the inclusion of references. The proper citation of references is considered to be the decent act of giving due credit to studies of theoretical foundations and important findings or disputes as to where certain information was retrieved from, all of which provide a solid academic background for the research in focus. Likewise, the number of references cited is regarded as a marker of the "scholarliness" of articles and journals [1] because it represents the extent to which researchers depend upon and are indebted to the knowledge and discoveries of others.

As writers, we suggest that the number of references cited in a paper also implies the amount of time and energy spent in reviewing previous work during the process of conducting experiments, analysis, and writing. It has been widely discussed that some measurable indexes

**Funding:** Funding for this research is provided by the Natural Science Foundation of Hubei Province of China grant 2019CFA066 (CD), National Science & Technology Fundamental Resources Investigation Program of China grant 2019FY101800 (CD), and the Innovation of Science and Technology Commission of Shenzhen grants JSGG20140515164852417 & JCYJ201206151530054 (TW).

**Competing interests:** The authors have declared that no competing interests exist.

of a research article, such as length (number of pages), capacity (bytes), and content (number of figures and tables), convey the energetic cost for writers and reviewers in the production of a publication (see, for example, [2]). References cited in an article, a great portion of which require careful intellectual examination, should be no exception. In addition, the above-mentioned aspects of a research article are typically correlated [2–4], suggesting the consistency and single-minded devotion that authors give to their writing endeavors.

Studies have shown a trend for increasing numbers of references across different time windows in the fields of psychology [5], biochemistry [6, 7], life sciences [2], polymer materials [8], and engineering [1]. Although a few studies have conducted multidisciplinary examinations [9–11], it is still not clear how citing behavior changes over longer periods of time, or what the current citing status is in major disciplines. Some researchers have proposed that the increase in the number of references is due to the increase in the total number of publications [3, 6, 10] and easy access to the literature since the year 2000 as the Internet and online academic publishing platforms become increasingly accessible [1, 5, 11], referred to hereafter as "millennium acceleration." However, we are not aware of any study that has compiled a large dataset and conducted a thorough analysis to empirically test these hypotheses. To this end, the aims of the current study are: (i) to investigate how the number of references per article has changed across several major academic disciplines between 1980 and 2019 and (ii) to test the two hypotheses proposed to explain the growth in the number of references.

## Materials and methods

Journal Citation Reports (JCR), included in Web of Science maintained by Clarivate Analytics®, was used between July and October, 2020 to retrieve journal and reference information. As far as we are concerned, the InCites JCR dataset was updated on Jun 29, 2020 and Oct 20, 2020. Thus, the data used in this study might not reflect the revisions made in October. However, we tried our best to double-check the consistency of key information—if contradictions occurred, updates were made or noted. In total, there were 236 disciplinary categories ranked by number of journals (JCR Year 2019 selected and both SCIE and SSCI edition ticked). Primarily, we chose the top 30 categories (see Table 1), the first being Economics, comprising 373 journals from the SSCI edition and the 29th and 30th tied between Biotechnology & Applied Microbiology and Computer Science, Information Systems, each composed of 156 journals from the SCIE edition. In total, the chosen subjects comprised 30/236 = 12.71% of all categories, and roughly 33.58% of all journals (6640/19 775, some overlaps may have occurred among the different categories).

Within each category, we used aggregate source data and retrieved the article number in the JCR Year (A), the number of references (B), and the ratio (B/A) for all years (starting in 2003 and ending in 2019). The only exception was that the ratio for articles in the category of Linguistics was only available starting in 2006. The ratios represent the average number of references cited by each article within a certain discipline in a given year (aka reference density).

To extend our study to a wider time frame (as early as 1980), eight categories from the top 30 were sampled in order to perform a more detailed investigation. Three principles guided the selection process: (i) the regression slopes for each level ($<0.7$, $0.7$–$1.4$, $>1.4$) should be represented; (ii) the categories of both the SCIE and SSCI editions should be included; and (iii) the selected categories should represent different essential disciplines (for instance, Mathematics and Mathematics, Applied could not be nominated simultaneously). Thus, the selected eight categories were: Mathematics; Education & Educational Research; Management; Geosciences, Multidisciplinary; Cell Biology; Linguistics; Ecology; and Computer Science, Information Systems.

**Table 1. Reference density (ratio) in 2003 (start point) and 2019 (end point) of the top 30 disciplinary categories included in Journal Citation Reports for year 2019.**

| Rank | Category | Edition | No. of journals | 2003 ratio | 2019 ratio | Slope* | $R^2$ |
|------|----------|---------|-----------------|------------|------------|--------|-------|
| 1 | Economics | SSCI | 373 | 28.1 | 47.5 | 1.22 | 97.9% |
| 2 | Mathematics | SCIE | 325 | 16.1 | 25.1 | 0.52 | 99.4% |
| 3 | Materials Science, Multidisciplinary | SCIE | 314 | 19 | 42.4 | 1.53 | 98.6% |
| 4 | Biochemistry & Molecular Biology | SCIE | 297 | 38 | 49.5 | 0.75 | 97.4% |
| 5 | Neurosciences | SCIE | 272 | 40.9 | 54.1 | 0.88 | 98.3% |
| 6 | Pharmacology & Pharmacy | SCIE | 271 | 29.7 | 39.8 | 0.78 | 85.9% |
| 7 | Engineering, Electrical & Electronic | SCIE | 266 | 16 | 34 | 1.09 | 97.7% |
| 8 | Environmental Sciences | SCIE | 265 | 29.9 | 53 | 1.48 | 98.8% |
| 9 | Education & Educational Research | SSCI | 263 | 30.5 | 51.8 | 1.32 | 99.6% |
| 10 | Mathematics, Applied | SCIE | 261 | 19.3 | 29.7 | 0.69 | 96.8% |
| 11 | Oncology | SCIE | 244 | 33.1 | 36.4 | 0.17 | 81.3% |
| 12 | Plant Sciences | SCIE | 234 | 33.9 | 51.8 | 1.16 | 99.6% |
| 13 | Management | SSCI | 226 | 37.9 | 70.7 | 2.24 | 97.8% |
| 14 | Surgery | SCIE | 210 | 21.9 | 27.6 | 0.39 | 97.6% |
| 15 | Clinical Neurology | SCIE | 204 | 29.8 | 36.1 | 0.50 | 95.8% |
| 16 | Geosciences, Multidisciplinary | SCIE | 200 | 26.1 | 59.4 | 1.93 | 96.6% |
| 17 | Cell Biology | SCIE | 195 | 40.3 | 50.3 | 0.63 | 97.3% |
| 18 | Public, Environmental & Occupational Health | SCIE | 193 | 28.3 | 39 | 0.60 | 97.0% |
| 19 | Linguistics | SSCI | 187 | 45.2^ | 54.1 | 0.78 | 91.8% |
| 20 | Political Science | SSCI | 181 | 25.6 | 54.8 | 1.97 | 97.0% |
| 21 | Genetics & Heredity | SCIE | 178 | 35.5 | 47 | 0.85 | 95.7% |
| 22 | Chemistry, Multidisciplinary | SCIE | 177 | 25.2 | 46.7 | 1.48 | 98.9% |
| 23 | Public, Environmental & Occupational Health | SSCI | 171 | 31.6 | 41.5 | 0.58 | 93.4% |
| 24 | Ecology | SCIE | 169 | 41.2 | 62 | 1.36 | 98.4% |
| 24 | Zoology | SCIE | 169 | 38.6 | 52 | 0.91 | 93.8% |
| 26 | Medicine, General & Internal | SCIE | 165 | 23.6 | 32.2 | 0.53 | 96.9% |
| 27 | Chemistry, Physical | SCIE | 159 | 29.4 | 49.7 | 1.36 | 98.9% |
| 27 | Immunology | SCIE | 159 | 32.6 | 41.7 | 0.59 | 98.6% |
| 29 | Biotechnology & Applied Microbiology | SCIE | 156 | 27.8 | 43.5 | 0.95 | 99.3% |
| 29 | Computer Science, Information Systems | SCIE | 156 | 23.2 | 39.3 | 1.08 | 98.9% |

Linear regression was conducted on the 17-year reference data for each category, and coefficients are reported. Rows shaded in gray are the categories selected for later investigation.

* All regression slopes are significant at $p < 0.0001$ level.

^ Reference data in Linguistics were only available starting from 2006.

In order to manually select articles to examine, journal selection was our primary concern. According to the default setting of JCR, journals within each category were ordered by their impact factors in 2019. Therefore, their ranks were used as unique journal IDs in our study. To randomly pick ten journals, ten random integers were generated with a limit of one to the total number of journals within a particular category. However, as we did not want to include journals that publish mainly review articles (only research articles were the target of our study) or have a very short publishing history (with a low potential for detecting trends), three criteria were used to aid journal selection: (i) the journal title or type must not indicate its publishing preference for reviews only; (ii) the publication frequency as indicated by JCR must be greater than four issues per year (as of 2019); and (iii) the year of a journal's inaugural issue must be

no later than 1998. Journals that met all three criteria were marked as "Yes," otherwise "No." If needed, more random integers were generated until the number of "Yes" journals reached a total of ten (see S1 Table in S1 File). Additionally, we checked the author guidelines of all "Yes" journals and ensured that there was no explicit requirement regarding the number of references cited.

For each selected journal (8 categories × 10 journals = 80 in total), we picked the median-numbered issue in each year to best represent each one. Ten research articles were randomly chosen from that issue (if less than ten, more issues were included or it was left as it was when no more issues could be found, especially for the early years). Article type was identified as "research" (original article, short report, letter, brief communication, but NOT review, commentary, opinion, perspective, editorial, etc.) typically based on the information found in electronic table of contents (eTOC) and article titles. We collected data including the year of publication, issue number, page range of a particular article (article length in pages), and total number of references cited. Each journal was traced back to 1980, or whenever the inaugural issue was released, up until 2002, after which, aggregate source data were available from JCR. For the category Linguistics, however, more data were collected for the years 2003–2005 due to lack of information from aggregate source data in JCR. The whole process yielded a total of 15 330 data entries. In order to double-check and spot outliers, histograms and plot figures were made for number of article pages, number of references, and number of references per page after all data were collected. Whenever outliers were found, the particular article was traced in order to examine it in detail. Besides typos, most outliers were due to unwanted article types, in which case, an appropriate substitution was made.

In addition, we checked the values of the ratio B/A for the years 2002–2019 for each journal selected and compared them with the category-level ratios for each year (one sample $t$-test, see S2 Table in S1 File). This acted as a two-way corroboration. If no difference was detected (which was the case for the majority of comparisons), we concluded that the ten selected journals well represented the category in the earlier years when aggregate source data were not available, as well as that the categorical aggregate values were in accordance with the sampled journals. Hence, we felt confident in combining data manually collected from journal articles for 1980–2002 and categorical data retrieved directly from JCR for 2003–2019.

Note that we use the terms "number of references per article" and the "ratio" or "reference density" [10] interchangeably in this paper. This refers to the number of cited papers in the bibliographic data at the single-paper level or the ratio of total references in a JCR year divided by the number of citable articles in each category.

## Statistical analysis

All statistical analyses and figures were done using R version 4.0.2 [12] and packages including car [13], psych [14], lmerTest [15], lsmeans [16], lme4 [17], and plotrix [18].

For the top 30 categories, we used the year as the predictive variable in order to conduct linear regression on the reference density to detect if there was an increasing trend in the number of references cited per article between 2003 and 2019.

In the eight selected categories during the period 1980–2002, pairwise correlation was tested among year, number of references cited per article, and article page length, and both Pearson and Spearman correlation coefficients were calculated (S3 Table in S1 File). In light of the positive relationship between number of references and article page length in all categories, we adopted a new parameter, references per page (defined as the number of references cited in an article divided by the number of pages of the article). In doing so, we were able to quantitatively investigate how the number of references changed with time, independent of paper

length, which was also assumed to increase with time [2] and indeed was found to be so in our dataset (S3 Table in S1 File). The trend in citing behavior between 1980 and 2002 was then analyzed using linear mixed-effects models. For each category, references per page was the dependent variable and year the continuous predictive variable. Because the reference data for a certain category were collected from ten random journals, each of which may have retained some consistency in journal-specific preferences through time, thus journal identity was included as the random effect. Significance level was estimated using Satterthwaite's method, implemented in package lmerTest::anova.

To examine the 40-year trend, due to the fact that there was only one data point per year for each category between 2003 and 2019, in order to lay balanced weight on the regression, median values from 1980–2002 was used to represent each year for each category (the parametric means were also tested and the results were similar, however, but with lower values of $R^2$). Hence, each category ended up with 40 data points across 40 years, which were analyzed using linear regression. In all models, diagnostic plots were constructed in order to ensure conformation with model assumptions of homoscedasticity and normal distribution of residuals.

In order to test if the increase in the number of references was affected by the increase in the number of publications in the top 30 categories, we calculated the average annual growth rate in the number of citable articles and reference density using the equations below (adopted from [10]).

$$\text{Annual growth rate} = [(\text{Number in 2019/Number in 2003})^{1/16} - 1] \times 100\%$$

A pairwise $t$-test was employed to compare the annual growth rate of journal article publications and reference density. The Pearson correlation coefficient was also calculated.

For the purpose of testing the millennium conjecture, the 40-year dataset was separated into two equal halves, one for 1980–1999, as the part prior to the millennium, and the other for 2000–2019, as the post-millennium part. Within each part, the numbers 1 to 20 were assigned to each reference density from the year of beginning to the end, respectively. Analysis of covariance (ANCOVA) was then employed to test the effect of time (continuous 1–20), millennium (pre- or post-), and their interaction for each disciplinary category. If the interaction term showed up as a significant source of variation, it implied that the rate of increase in the number of references per article differed between the first and second 20-year periods, thus supporting the conjecture of post-millennium acceleration.

## Results

### Increase in number of references per research article with time

Among the top 30 academic disciplinary categories, all showed significant increasing trends in the number of references cited per research article from 2003 to 2019, with high levels of $r$-squared values, ranging from 81.3% to 99.6% (Table 1). However, the rate of increase varied among the categories. The lowest rate of increase was only 10% in Oncology, while the highest rates more than doubled the reference density in 2019 compared to 2003 (Materials Science, Engineering, Geosciences, and Political Science). On average, the number of references cited per article was 29 in 2003, which increased to 45 in 2019.

For the period 1980–2002, not only did the number of references per article show an increasing trend as the years progressed in all eight chosen categories (S3 Table in S1 File), but also the number of references per page, after subtracting the confounding factor of paper length, demonstrated a significant growing trend (Fig 1, Table 2). The only exception was Mathematics, which overall had low reference counts and was almost flat across all years. Among all documented data, the maximum reference counts per article and per page were 58

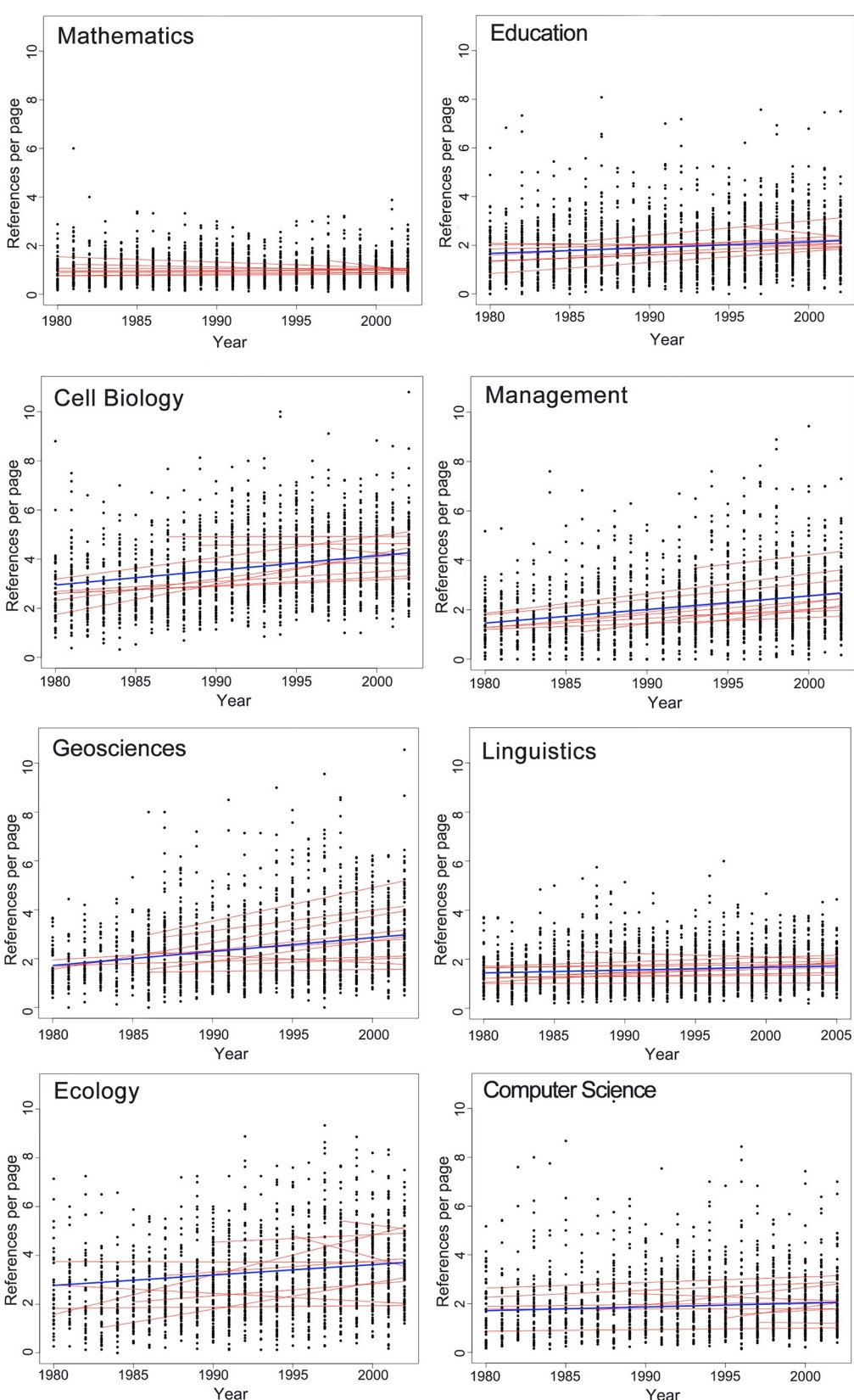

**Fig 1. Changes in the number of references cited per page in eight disciplinary categories from 1980 to 2002 (1980–2005 for Linguistics).** The blue lines represent categorical linear regression (only significant lines drawn), while the thin red lines represent the specific performance of each journal.

and 6, respectively, for Mathematics, 130 and 8.08 for Education, 153 and 9.43 for Management, 144 and 10.55 for Geosciences, 116 and 10.8 for Cell Biology, 113 and 6 for Linguistics, 100 and 9.33 for Ecology, and 123 and 10.29 for Computer Science.

Between 1980 and 2019, all eight categories showed significantly increasing trends in the number of references per article (Fig 2, Table 3). As of 2019, Mathematics had the lowest number of average reference counts, corresponding to the smallest growth rate, while Management had the highest number of reference counts, corresponding to the steepest rate of increase.

## Conjectures as to publication growth and millennium acceleration

As to the speculation regarding the increase in the level of publication generally, the number of citable articles showed an enormous rate of increase from 2003 to 2019, with the lowest annual growth rate of 1.49% in Biochemistry and Molecular Biology and the highest growth rate of 13.18% in Computer Science (Table 4). At the same time, the annual growth rate in reference density in most categories showed a moderate level compared to the corresponding bursts in the number of articles (paired-t = 6.54, $p < 0.0001$). Among the 30 selected academic disciplinary categories, the annual growth rate of citable articles and reference density were positively correlated ($r = 0.37$, $p = 0.047$), implying that the increase in the number of publications promoted reference citation. However, in categories of Biochemistry and Geosciences, the rates of increase in reference density exceeded those of articles respectively, suggesting that the number of references might increase faster than the number of articles in certain fields.

When the study time frame was divided into two equal 20-year periods (1980–1999 and 2000–2019), all categories showed the same pattern, namely, that the second period had an overall higher number of references than the first, and both increased year on year, which can easily be seen from Fig 2. However, as for the interaction term, there was no significant pattern of acceleration in the growth rate of reference counts for the period following the millennium in the categories Education ($F_{1,36} = 0.66$, $p = 0.42$), Cell Biology ($F_{1,36} = 2.89$, $p = 0.098$), and Linguistics ($F_{1,36} = 1.11$, $p = 0.30$), whereas the conjecture regarding millennium acceleration was supported in the categories Mathematics ($F_{1,36} = 28.75$, $p < 0.0001$), Management ($F_{1,36} = 41.88$, $p < 0.0001$), Geosciences ($F_{1,36} = 11.29$, $p = 0.0019$), Ecology ($F_{1,36} = 12.76$, $p = 0.001$), and Computer Science ($F_{1,36} = 24.38$, $p < 0.0001$).

**Table 2. Statistical results for linear mixed-effects models testing the increasing trend in the number of references per page in eight disciplinary categories from 1980 to 2002.**

| Category | Linear coefficient of Year | DF (denominator) | F value | p |
|---|---|---|---|---|
| Mathematics | −0.0022 | 2010 | 1.39 | 0.24 |
| Education | 0.024 | 2026.9 | 39.85 | <0.0001 |
| Management | 0.055 | 1797.2 | 135.2 | <0.0001 |
| Geosciences | 0.057 | 1863.4 | 140.13 | <0.0001 |
| Cell Biology | 0.060 | 1867.8 | 162.2 | <0.0001 |
| Linguistics | 0.011 | 2352.6 | 28.88 | <0.0001 |
| Ecology | 0.043 | 1716 | 58.87 | <0.0001 |
| Computer Science | 0.015 | 1657.2 | 10.74 | 0.0011 |

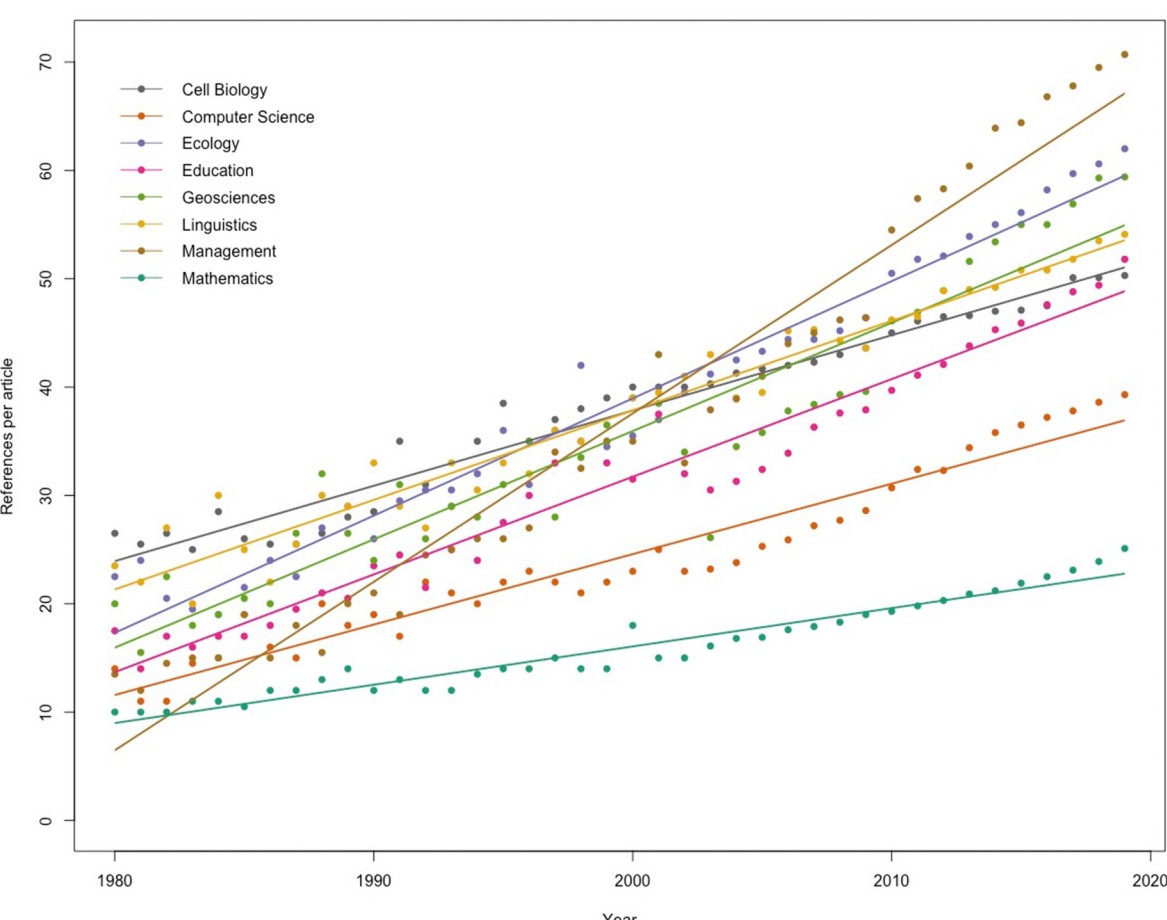

**Fig 2. Linear trends in the number of references cited per research article from 1980 to 2019 in eight disciplinary categories.**

**Table 3. Statistical results for the 40-year (1980–2019) linear trend in the number of references cited per research article in eight disciplinary categories.**

| Category | Slope | F value* | $R^2$ |
|---|---|---|---|
| Mathematics | 0.35 | 625.52 | 94.3% |
| Education | 0.90 | 829.63 | 95.6% |
| Management | 1.56 | 916.97 | 96.0% |
| Geosciences | 1.00 | 318.74 | 89.4% |
| Cell Biology | 0.70 | 806.61 | 95.5% |
| Linguistics | 0.83 | 780.20 | 95.4% |
| Ecology | 1.08 | 1128.40 | 96.7% |
| Computer Science | 0.65 | 551.52 | 93.6% |

* All F values correspond to a significance level of $p < 0.0001$.

**Table 4. Annual growth rate of citable articles from 2003 to 2019 in the top 30 disciplinary categories compared to that of reference density.**

| Category* | Number of citable articles | | Annual growth rate (%) | |
|---|---|---|---|---|
| | Year 2003 | Year 2019 | Article | Reference density |
| Economics | 7085 | 21280 | 7.12 | 3.34 |
| Mathematics | 12364 | 28969 | 5.47 | 2.81 |
| Materials Science | 29249 | 124929 | 9.50 | 5.14 |
| Biochemistry | 42730 | 54107 | 1.49 | 1.67 |
| Neurosciences | 22280 | 34625 | 2.79 | 1.76 |
| Pharmacology | 20071 | 36728 | 3.85 | 1.86 |
| Engineering | 25524 | 83047 | 7.65 | 4.82 |
| Environmental Sciences | 15388 | 70505 | 9.98 | 3.64 |
| Education | 2866 | 12143 | 9.44 | 3.37 |
| Mathematics, Applied | 13301 | 26900 | 4.50 | 2.73 |
| Oncology | 17221 | 40599 | 5.51 | 0.60 |
| Plant Sciences | 12601 | 24246 | 4.18 | 2.69 |
| Management | 2601 | 11239 | 9.58 | 3.97 |
| Surgery | 21152 | 33551 | 2.93 | 1.46 |
| Clinical Neurology | 13522 | 25857 | 4.13 | 1.21 |
| Geosciences | 12187 | 26060 | 4.86 | 5.27 |
| Cell Biology | 15906 | 27564 | 3.50 | 1.39 |
| Public Health (SCIE) | 7740 | 25750 | 7.80 | 2.02 |
| Linguistics | 1250^ | 5586 | 12.21 | 1.39 |
| Political Science | 2869 | 7040 | 5.77 | 4.87 |
| Genetics | 12310 | 20483 | 3.23 | 1.77 |
| Chemistry | 21378 | 75601 | 8.21 | 3.93 |
| Public Health (SSCI) | 3356 | 20199 | 11.87 | 1.72 |
| Ecology | 9313 | 18901 | 4.52 | 2.59 |
| Zoology | 6082 | 11345 | 3.97 | 1.88 |
| Medicine, General | 10725 | 26894 | 5.91 | 1.96 |
| Chemistry, Physical | 23582 | 72148 | 7.24 | 3.34 |
| Immunology | 15260 | 20283 | 1.79 | 1.55 |
| Biotechnology | 13228 | 24438 | 3.91 | 2.84 |
| Computer Science | 4709 | 34127 | 13.18 | 3.35 |

* Refer to Table 1 for the list of full names of all categories.

^ Data for the number of citable articles in Linguistics were only available starting from 2006, hence the reference density. The equations for calculating the annual growth rate were adjusted accordingly.

## Discussion

A central theme of the results is that the number of references cited in research papers has been increasing, both per article and per page, over time frames of 17, 23, and 40 years, among the 30 selected disciplinary categories or the 80 journals of the selected categories in the natural and social sciences. Such a growing trend is so robust that most linear regression on year explained more than 90% of the variation in the number of references cited, although there were noticeable differences in the intensity of growth among some disciplines. Our results showed comparable reference counts with those of Biglu [9], who collected a large dataset from multiple disciplines at five-year intervals. Additionally, the increasing trend also linked very nicely with studies reporting 6–8 reference counts per article in the 1950s, and

approximately 15 in the 1960s and 1970s (reviewed in [7] and references therein), together presenting a steady increase in the number of references with time.

Will the increase in reference numbers ever level off? Before we can make a realistic prediction, it might be enlightening to review what researchers have foreseen in the past. Meadows in 1974 assumed that most increases seemed to slow down after approximately 15 references per paper [7]. Price in 1965 also stated that the norm of scholarship is a paper with proximately 10–22 references [7], the upper limit of which had been surpassed by the average reference densities of eight disciplines in 1988 (Fig 2), and by that of Mathematics, with the lowest reference count among the 30 selected categories, in 2016. From a perspective of gathering more citations, Lovaglia [19] estimated that the ideal number of references for a sociological journal article is 66, which had already been met by the reference density of Management in 2016 (Fig 2). Nevertheless, it did not show any sign of leveling off, yet continued to grow yearly, reaching 71 in 2019. Yitzhaki & Ben-Tamar [7] predicted that most papers would have up to 50 references in 1990 and that papers with more than 51 references would continue to grow till 2000. However, researchers also anticipated that, at some point, the increasing trend would slow down and enter a saturation phase [6, 7], otherwise "all papers will become reviews"! Yet, the inflection point was not explicitly specified. For now, we are sure that, for most disciplines, the inflection point did not occur in 2005 [9], nor in 2013 [1], nor in 2015 [10], and not even by 2019 (shown in Fig 2, also in [11]). It appears that the reference lists of journal articles have undergone a runaway process of ever-expanding increase.

Despite the general rise in reference density, the growth pattern showed great disparities among the various disciplines. Mathematics performed remarkably, as it showed the lowest reference density and the flattest growth between 1980 and 2019. Such results matched closely with studies that found habitually lower reference counts per journal article in mathematics, engineering, and statistics [1, 7, 10], perhaps due to the intrinsic nature of math- and technology-related fields. Publications in relation to Mathematics, as well as Linguistics, are also considered problematic in bibliometric studies because they may often cite or be cited from sources not covered by Web of Science, especially in early years [20]. In contrast, Management showed the steepest growth pattern and the highest reference density as of 2019 among the 30 selected categories. Likewise, Business, Management, and Accounting was found to be the fastest growing category among the social sciences, probably owing to its lower reference counts in earlier years, "as if these scientific fields would try to make up for lost time" [10]. Furthermore, biomedical and medicine-related categories (i.e., ranked 4, 5, 11, 14, 15, 26, and 27 in Table 1) behaved quite similarly, with medium to low rates of increase, suggesting a limited level of increase in reference density. Sánchez-Gil *et al*. [10] also hinted at such a phenomenon in Health Science in general. Although we are unsure what the exact reasons for this constrained growth in reference density in these fields are, scattered evidence suggests that guidelines as to upper limits for the number of references in medical journals might have long been established ([21], and information on journal websites).

Some argue that when the pool of literature increases year on year, it is inevitable that articles will cite ever greater numbers of references [6, 10]. Our results agree with this hypothesis in that, among the 30 selected subject categories, a higher annual growth rate in journal articles is associated with a higher annual growth rate in reference density. However, we think that it would be an oversimplification to assume that the reason for growth in the number of references is due to an increase in the overall number of published articles. For one thing, the two have distinct properties. While the number of yearly publications represents an aggregate, the number of references per article is an attribute of a single scientific document, which has its internal rules and limits. Secondly, when we examined the data by category in detail, we found that the top three growth rates in numbers of journal articles (i.e., Computer Science,

Linguistics, and Public Health) were not linked with the top three growth rates in reference density (i.e., Geosciences, Materials Science, and Political Science). Such mismatches are also reflected by a moderately low correlation coefficient (0.37, although significant). Similarly, Sánchez-Gil *et al.* [10], using different data sources, failed to detect any correspondence between the two types of growth rates. Following the same train of thought, the conjecture of millennium acceleration hints at authors' greater ability to access more digitized literature around or after the year 2000 [1]. Our results only partially supported this speculation, however. Five out of eight categories indeed showed steeper growth with time in respect of the number of references per article during 2000–2019 than during 1980–1999. Interestingly, studies have found that older papers (older than five years) comprise a higher percentage in the reference lists of newer journal articles [5, 6], suggesting the rediscovery of older papers rather than intensive citation of contemporary ones, which refutes both conjectures. To conclude, albeit the two conjectures may provide somewhat sensible explanations in certain fields, they do not appear strong enough to ascribe the reason for the growth in reference density to an overall increase in publication or to easier digital access to the literature.

Many researchers believe that the change in citing behavior, leading to increasingly longer lists of references, is deeply rooted in the perceived benefit of gathering future citations [4, 9–10, 19, 22]. As impact factors and performance indicators have long been an intrinsic part of academic publishing [23], academics have become ever more desirous to publish in prominent journals and to be cited more often. Having more references in their papers becomes a handy style because it not only allows greater visibility in literature networks (i.e., citation consciousness [6, 9]) but also engenders more academic connections that encourage future mutual citation (i.e., tit-for-tat [4, 22]). However, naïve writers may feel obligated to review much of the literature and have long reference lists simply because the authors of the papers they read do the same (i.e., herd behavior [2]). Proactively or grudgingly, authors have made a collective effort to raise the bar for the number of cited references ever higher over time, which has ever-increasing consequences (Fig 2). In our opinion, despite growing numbers of references being fashionable, or even politically advantageous, the fundamental rules of academic paper writing should always be respected and observed: to list only relevant and significant references [24].

It is worth mentioning that we have not exhausted our examination of all of the possible reasons for the increasing trend in citing behavior, the foremost of which is the growing complexity of research itself [2, 4]. Since all academic fields are undergoing ever-increasing expansion, it is unavoidable that academics will postulate ever more theories and models, use more materials, technologies, and methodologies, and observe more phenomena, not to mention having to formulate and deal with the increasingly intricate linkages among them. As a result, more and more references are needed to ensure the adequacy of literature review. Nevertheless, it has been asserted that a single academic paper, highlighting one or perhaps a few points of originality (as it always should be), should not require an ever-broader foundation of knowledge as the years progress [1]. Another related factor is the length of papers, which is thought to be an important determinant of the number of references (see, for example, [3]). Although we have not surveyed paper length over the 40-year period we studied, based on our analysis of the number of references per page during 1980–2002 (Fig 1), it is highly likely that there continues to be an increasing trend independent of paper length. Such a conclusion was also reached by Ucar *et al.* [1], who revealed a page-level trend until 2013. In addition, researchers have speculated about other factors, including the advancement of teamwork and multiple authorships [5], pressure from reviewers [2], and the preferences of journal editors and readers [19]. Another limitation of this study is that our data come from a single source, namely, Journal Citation Reports, as opposed to others that consider Scopus, which is thought to have a wider coverage [10, 11].

In summary, our investigation revealed the fact that reference lists have continued to lengthen in the journal articles of many fields between 1980 and 2019. The yearly expansion in academic publications and greater accessibility may partly account for the increase in reference density. If this trend were to continue, would most articles effectively become reviews? Would the reference sections become longer than the main body of journal articles? Would younger scientists wishing to publish their work be intimidated? We feel that it is time to stop making the communication of science unnecessarily demanding. It is we, the existing contributors to academic publishing, who must act to break this runaway trend. A good starting point, perhaps, is to cite less. As quoted by Gastel and Day [24]: "Manuscripts containing innumerable references are more likely a sign of insecurity than a mark of scholarship (William C. Roberts)". We hope that all researchers will have the faith and freedom to recognize their significant references and to keep their reference lists as short (or long) as they need to be.

## Supporting information

**S1 File.**
(DOCX)

## Acknowledgments

We are greatly indebted to P. Zhou, K. Niu, X. Xie, X. Chen, Y. Wu and J. Xie for their help in data collection.

## Author Contributions

**Conceptualization:** Can Dai, Tao Wan, Fan Liu, Yanbing Gong, Qingfeng Wang.

**Funding acquisition:** Can Dai, Tao Wan, Qingfeng Wang.

**Investigation:** Can Dai, Quan Chen.

**Methodology:** Can Dai, Tao Wan, Qingfeng Wang.

**Project administration:** Qingfeng Wang.

**Visualization:** Can Dai, Quan Chen, Tao Wan.

**Writing – original draft:** Can Dai, Fan Liu, Yanbing Gong.

**Writing – review & editing:** Quan Chen, Tao Wan, Qingfeng Wang.

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
