## [Decision Letter · Decision Letter 0]

16 Jun 2021

PONE-D-21-13673

Literary runaway: Increasingly more references cited per academic research article from 1980 to 2019

PLOS ONE

Dear Dr. Wang,

Thank you for submitting your manuscript to PLOS ONE. After careful consideration, we feel that it has merit but does not fully meet PLOS ONE’s publication criteria as it currently stands. Therefore, we invite you to submit a revised version of the manuscript that addresses the points raised during the review process.

We look forward to receiving your revised manuscript.

Kind regards,

Roy Cerqueti, Ph.D.

Academic Editor

PLOS ONE

Journal Requirements:

3. Please remove your figures from within your manuscript file, leaving only the individual TIFF/EPS image files, uploaded separately.  These will be automatically included in the reviewers’ PDF.

Additional Editor Comments (if provided):

Reviewers' comments:

Reviewer's Responses to Questions

**Comments to the Author**

1. Is the manuscript technically sound, and do the data support the conclusions?

Reviewer #1: Yes

Reviewer #2: Yes

2. Has the statistical analysis been performed appropriately and rigorously? 

Reviewer #1: Yes

Reviewer #2: Yes

3. Have the authors made all data underlying the findings in their manuscript fully available?

Reviewer #1: Yes

Reviewer #2: No

4. Is the manuscript presented in an intelligible fashion and written in standard English?

Reviewer #1: Yes

Reviewer #2: Yes

5. Review Comments to the Author

Reviewer #1: I appreciate the work done by all the autors. I do not find any short comings in the manuscript. First of all, the literature study is comprehensive and it covers the said topic. The language used is appropriate and upto the mark.

Reviewer #2: The manuscript "Literary runaway: Increasingly more references cited per academic research article from 1980 to 2019" presents an analysis of the average number of references in scientific citable papers. Eight subject categories are sampled for the time period 1980-2003 and thirty subject categories are analyzed for two years (2003 and 2019). Statistical analyses show an overall increasing number of references per paper on average over the years. Overall, the manuscript is well written and should be of interest to the readership of PLoS One. However, some improvements should be made before I can recommend publication of this manuscript.

The data are analyzed using R, and the authors pledge to make the data available upon acceptance. For the increasing benefit of the readers and to help reproducing the results, I would appreciate if the authors would share their R code with the readers.

I wonder if the authors could provide confidence intervals in the plots of Fig. 2.

Two of the sampled subject categories are Mathematics and Linguistics. These subject categories should be regarded as problematic for bibliometric treatment. A discussion about this is warranted.

The authors analyze all cited references and also cite previous work that analyzed all cited references in papers as far as I can see. DOI 10.1016/j.joi.2016.07.002 also investigated the growth of linked cited references within a three-year window. This might be interesting in this context as it shows that the increasing bulk of previous knowledge plus the recent knowledge is not the reason for the observed trend. Only the references to the recent knowledge are increasing on average over time, too.

Did the authors check if the average number of references is increasing for it is corrected for the increasing number of papers per year?

The last paragraph of the Discussion section in lines 400-402 starts with an irritating sentence: "In summary, we conducted this research in order to raise awareness about the fact that reference lists have continued to lengthen in the journal articles of many fields between 1980 and 2019." In the Introduction section, the authors mentioned selected previous studies that already raised the awareness to this fact. Furthermore, the new contribution of this study is testing of the two hypotheses that are mentioned in the end of the Introduction section.

6. PLOS authors have the option to publish the peer review history of their article (what does this mean?). If published, this will include your full peer review and any attached files.

Reviewer #1: No

Reviewer #2: No

---

## [Author Response · Author response to Decision Letter 0]

19 Jun 2021

Please see the uploaded document

---

## [Decision Letter · Decision Letter 1]

26 Jul 2021

Literary runaway: Increasingly more references cited per academic research article from 1980 to 2019

PONE-D-21-13673R1

Dear Dr. Wang,

We’re pleased to inform you that your manuscript has been judged scientifically suitable for publication and will be formally accepted for publication once it meets all outstanding technical requirements.

Kind regards,

Roy Cerqueti, Ph.D.

Academic Editor

PLOS ONE

Additional Editor Comments (optional):

Reviewers' comments:

Reviewer's Responses to Questions

**Comments to the Author**

1. If the authors have adequately addressed your comments raised in a previous round of review and you feel that this manuscript is now acceptable for publication, you may indicate that here to bypass the “Comments to the Author” section, enter your conflict of interest statement in the “Confidential to Editor” section, and submit your "Accept" recommendation.

Reviewer #2: All comments have been addressed

2. Is the manuscript technically sound, and do the data support the conclusions?

Reviewer #2: Yes

3. Has the statistical analysis been performed appropriately and rigorously? 

Reviewer #2: Yes

4. Have the authors made all data underlying the findings in their manuscript fully available?

Reviewer #2: Yes

5. Is the manuscript presented in an intelligible fashion and written in standard English?

Reviewer #2: Yes

6. Review Comments to the Author

Reviewer #2: (No Response)

7. PLOS authors have the option to publish the peer review history of their article (what does this mean?). If published, this will include your full peer review and any attached files.

Reviewer #2: No

---

## [Editor Report · Acceptance letter]

29 Jul 2021

PONE-D-21-13673R1 

Literary runaway: Increasingly more references cited per academic research article from 1980 to 2019 

Dear Dr. Wang:

I'm pleased to inform you that your manuscript has been deemed suitable for publication in PLOS ONE. Congratulations! Your manuscript is now with our production department. 

Kind regards, 

on behalf of

Professor Roy Cerqueti 

Academic Editor

PLOS ONE